# Divergent and Convergent Attitudes and Views of General Practitioners and Community Pharmacists to Collaboratively Implement Antimicrobial Stewardship Programs in Australia: A Nationwide Study

**DOI:** 10.3390/antibiotics10010047

**Published:** 2021-01-05

**Authors:** Sajal K. Saha, David C. M. Kong, Karin Thursky, Danielle Mazza

**Affiliations:** 1Department of General Practice, Monash University, Building 1, 270 Ferntree Gully Road, Notting Hill, VIC 3168, Australia; Danielle.Mazza@monash.edu; 2National Centre for Antimicrobial Stewardship (NCAS), The Peter Doherty Institute for Infection and Immunity, Melbourne, VIC 3000, Australia; david.kong@monash.edu (D.C.M.K.); karin.thursky@mh.org.au (K.T.); 3Centre for Medicine Use and Safety, Monash University, 381 Royal Parade, Parkville, VIC 3052, Australia; 4Department of Medicine, University of Melbourne, Melbourne, VIC 3010, Australia; 5Pharmacy Department, Ballarat Health Services, Ballarat, VIC 3350, Australia

**Keywords:** antimicrobial stewardship, GP–pharmacist collaboration, GPPAS model, attitudes, views, general practitioners, community pharmacists

## Abstract

Setting up an interprofessional team for antimicrobial stewardship (AMS) to improve the quality and safety of antimicrobial use in primary care is essential but challenging. This study aimed to investigate the convergent and divergent attitudes and views of general practitioners (GPs) and community pharmacists (CPs) about AMS implementation and their perceived challenges of collaboration to design a GP–pharmacist collaborative AMS (GPPAS) model. Nationwide surveys of GPs and CPs across Australia were conducted January-October 2019. Chi square statistics and a theoretical framework were used for comparative analyses of quantitative and qualitative data, respectively. In total, 999 participants responded to the surveys with 15.4% (*n* = 386) response rates for GPs and 30.7% (*n* = 613) for CPs. GPs and CPs were aware about AMS however their interprofessional perceptions varied to the benefits of AMS programs. CPs indicated that they would need AMS training; significantly higher than GPs (GP vs. CP; 46.4% vs. 76.5%; *p* < 0.0001). GPs’ use of the Therapeutic Guideline Antibiotic was much higher than CPs (83.2% vs. 45.5%; *p* < 0.0001). No interprofessional difference was found in the very-limited use of patient information leaflets (*p* < 0.1162) and point-of-care tests (*p* < 0.7848). While CPs were more willing (*p* < 0.0001) to collaborate with GPs, both groups were convergent in views that policies that support GP–CP collaboration are needed to implement GPPAS strategies. GP–pharmacist collaborative group meetings (54.9% vs. 82.5%) and antimicrobial audit (46.1% vs. 86.5%) models were inter-professionally supported to optimise antimicrobial therapy, but an attitudinal divergence was significant (*p* < 0.001). The challenges towards GP–CP collaboration in AMS were identified by both at personal, logistical and organisational environment level. There are opportunities for GP–CP collaboration to improve AMS in Australian primary care. However, strengthening GP–pharmacy collaborative system structure and practice agreements is a priority to improve interprofessional trust, competencies, and communications for AMS and to establish a GPPAS model in future.

## 1. Introduction

Primary care significantly contributes to the rising antimicrobial resistance (AMR) because both prescribing and inappropriate prescribing of antimicrobials occur most in this setting. [1] However, in response, the actions and framework to implement antimicrobial stewardship (AMS) programs with an interprofessional engagement are limited in this setting. [2] AMS programs are strategic interventions that aim to optimise antimicrobial use, improve patient safety, and reduce AMR [3].

There are international guidelines and recommendations, practical tool kits, and models of care to facilitate AMS programs in primary care [4,5,6,7] but, attempts to implement these in clinical practice have significant challenges [8]. A key challenge is the lack of team-based antimicrobial care provision or services in primary care, where prescribing and dispensing occur independently and without pharmacists acting as provider of triage services. The prescribing etiquette and management of antimicrobial therapy is multi-disciplinary and is strongly influenced by interactions between the different health care professions [9].

General practitioners (GPs) and community pharmacists (CPs) are the most important stewards of antimicrobial use [10,11] in primary care, for which there should be a greater scope for collaboration between these two professions [12,13] to implement effective AMS programs. CPs can act as the gateway practitioners and the sources of antimicrobial information for both patients and GPs. Thus, CPs can assist patients in avoiding unnecessary visits to GPs and putting pressure on GPs to prescribe antimicrobials where it is not warranted. However, practical GP–CP collaborative models for AMS in the Australian health care setting remain poorly delineated; context-specific challenges and effective attempts are neither well known nor clear. Although doctor-pharmacist collaborative models exist to promote AMS in a secondary or tertiary care settings [14,15], it may be unrealistic to use these models in primary care due to the divergence in routine practices, distant working environment, organisational set-up, and different patterns of antimicrobial use between health care settings.

The evidence is growing that GP–pharmacist team-based AMS interventions can optimise antimicrobial use in primary care but the scale of implementation is unclear. [16] Effective interventions include GP–pharmacist group meetings, post-prescription review and feedback, GP–pharmacy practice agreements, point-of-care tests using a collaborative service model, team-based academic detailing, and use of patient forwarding information leaflets [17,18,19,20]. However, the difficulties of GPs and CPs using these strategies in a collaborative fashion are contextual and remain less well understood. In Australia, there is a dearth of information around the attitudinal models of collaboration which describes GPs’ and CPs’ attitudes towards collaboration in optimising antimicrobial use. While there are a few GP–CP collaboration models [21,22] supporting chronic care in Australia, a GP–CP model which facilitates AMS implementation in primary care remains unexplored. 

This study compares and contrasts the attitudes and views of GPs and CPs about AMS programs, their use of GP–pharmacist collaborative AMS (GPPAS) strategies, and challenges of collaboration in AMS to better understand their convergent and divergent views regarding collaborative AMS implementation with an aim to develop a GPPAS model in Australian primary care in future.

## 2. Results

### 2.1. Survey Responses and Characteristics of Respondents

The response rate of GPs (15.4%, 386/2500) was lower than CPs (30.7%, 613/2000) though the proportionate distribution of responses of both GPs and CPs across six Australian states and two territories and by gender was similar to the reports published by the Australian Government Department of Health. [23] Among the respondents, 221 GPs and 592 CPs responded to the open-ended questions. The demographic characteristics of GPs and CPs are shown in Table 1.

Among key characteristics, most participating GPs (79%) and CPs (84.6%) hold MBBS (Bachelor of Medicine and Bachelor of Surgery) and B. Pharm (Bachelor of Pharmacy) degrees, respectively. Respondents who had more than ten years of practice or work experiences were more common among GPs (GPs vs. CPs; 83.4% vs. 50.6%). More CPs than GPs completed their professional training in Australia (GPs vs. CPs; 67.7% vs. 90.0%). More than two-thirds of participants did not complete the National Prescribing Services (NPS)-developed educational antimicrobial modules (GPs vs. CPs; 72.5% vs. 81%) that guide optimal use of antimicrobials.

### 2.2. Convergent and Divergent Views about AMS Programs, Their Adoption in Practice and Collaboration Attitudes

We compared quantitative responses of 386 GPs and 613 CPs to identify the convergent and divergent views regarding AMS programs and their implementation in collaboration. Table 2 provides the comparative results.

#### 2.2.1. Awareness of, and Perception about, AMS Programs

Although both GPs and CPs were familiar with AMS, their perception regarding AMS programs varied (Table 2). In contrast to GPs, more CPs (*p*< 0.0001) believed that AMS programs have the potential to reduce health care costs associated with infection. On the other hand, a misperception that individual efforts have minimal impact to reduce AMR, was significantly higher (*p* < 0.0065) among CPs than GPs. CPs felt the need for AMS training significantly more than GPs (GP vs. CP; 46.4% vs. 76.5%; *p* < 0.0001).

#### 2.2.2. AMS Activities

GPs’ use of Antibiotic guidelines (Therapeutic Guideline) [24] was much higher than CPs (GP vs. CP; 83.2% vs. 45.5%; *p* < 0.0001; Table 2). However, there was no interprofessional difference in the use of patient information leaflets (*p* < 0.116) and point-of-care tests (*p* < 0.784) during patient consultations. Less than 20% of both GPs and CPs used these strategies, demonstrating poor uptake. In contrast, most of the participants educated their patients about unintended consequences (e.g., AMR, impact on gut microbiota) of using antimicrobials but CPs educated more than GPs (*p* < 0.0353).

#### 2.2.3. Attitudes towards GP–Pharmacist Collaboration in AMS

Most CPs and the majority of GPs were found to be willing to collaborate on issues related to AMS; this collaboration attitudes though significantly higher (*p* < 0.0001) among CPs than GPs. Overall, CPs were more supportive to implement GPPAS strategies and introduce GP–pharmacist collaborative AMS models compared to GPs. A majority in both groups strongly believed that improving AMS will need a policy that supports better collaboration between the GPs and CPs. However, CPs more strongly perceived that GPs should be receptive to CPs’ recommendation on the choice (GP vs. CP; 50.5% vs. 92.6%; *p*< 0.0001) and dose (GP vs. CP; 63% vs. 93.6%; *p* < 0.0001) of antimicrobials. Both GPs and CPs were supportive of collaborative group meetings (GP vs. CP; 54.9% vs. 82.5%; *p*< 0.0001) to set AMS goals and improve AMS practices, but GP–CP attitudinal divergence was significant.

Both were convergent in a view that pharmacists who have expertise in antimicrobials and infections should have regular group meetings with GPs to share the knowledge of antimicrobial pharmacotherapy. Conversely, less than half of GPs showed an interest in a pharmacist being co-located in the general practice to facilitate optimising of antimicrobial therapy (GP vs. CP; 39.5% vs. 79.5%; *p* < 0.001). Furthermore, though CPs strongly believed that the use of ‘My Health Records’ (MHR) would benefit in improving the inter-professional communication about antimicrobial prescriptions, GPs hold less strong belief on this benefit; (GP vs. CP; 30.9% vs. 67.2%; *p* < 0.0001).

#### 2.2.4. Attitudes towards Future Needs to Improve AMS

Most of the participant GPs and CPs were willing to participate in future AMS training programs and to have standard clinical care guidelines that would guide AMS activities in general practice and community pharmacies. Both professions agreed for a policy that would restrict the access to selected antimicrobials at community level. While most GPs did not want professional organisation (e.g., Royal Australian College of General Practitioners, RACGP) to define their AMS roles, CPs more strongly supported (*p* < 0.0001) their roles to be defined by their professional organisation (e.g., Pharmaceutical Society of Australia). We found significant discordance (*p* < 0.0001) in the attitudes towards involving CPs in implementing audit-feedback AMS strategies. Whereas 86% of CPs supported this involvement, less than half of GPs agreed [54% of GPs disagreed or neutral].

### 2.3. Barriers to Collaboratively Implement AMS Programs

According to our free-text analyses, we were able to identify the major barriers to GP–CP collaboration in matters related to AMS at person level, tools and technology level, interprofessional interaction level and environment level. The percentage of GPs and CPs whose qualitative responses fell into those different levels (themes) are demonstrated in Table 3. Of GPs and CPs who reported barriers under each level, percentage of them fell into categories of subthemes were also calculated and demonstrated in Table 3. The content analysis results showed that personal level barriers were reported by highest percentage of GPs and CPs with 44.8% and 56.4% respectively followed by environment, interaction and tools and technology level. The representative quotes under each subtheme are shown in Table 3.

#### 2.3.1. Professional Training in AMS Courses

The most common substantial barrier for both GPs and CPs was a lack of professional training in antimicrobial prescribing. GPs reported their poor undergraduate and postgraduate training in antimicrobial pharmacology whereas CPs highlighted that they were not keeping up to date on clinical guidelines, point-of-care diagnostics, and the latest trends in AMR. Hence, they might be unaware of their individual roles and collective actions while consulting or advising a patient with an infection. CPs stressed more on the lack of availability of protocols that would clearly guide routine AMS practice including interprofessional activities and checklists that might be used during patient consultation. Importantly, most of GPs and CPs were willing to receive AMS training if it was made available in future.

#### 2.3.2. Interprofessional Trust

A lack of trust towards each other’s professional competency emerged as another potential barrier for collaboration. Some GPs perceived that CPs’ skills were only related to dispensing practices and some GPs would find it offensive if CPs want to provide advice about antimicrobial prescribing. Consistently, most CPs reported that GPs neither do believe nor have confidence in their knowledge or ability to advise GPs on antimicrobial therapy. This indicates that there is a gap in promoting inter-professional education, the defined roles and dependency of each profession in improving the uptake of AMS strategies in primary care. Both professions stated that each other’s familiarity, long-term relationship, and self-motivation may increase trust and ease collaboration on matters related to AMS.

#### 2.3.3. Resistance

Both GPs and CPs reported that GPs sometimes feel criticised when CPs challenge their prescribing and sometimes the GPs would feel that CPs were overstepping their boundary or justification. CPs hold a perception that some GPs might not accept their recommendations for changing the decision of antimicrobials after the GPs have already prescribed the antimicrobial therapy. This may partly explain why some CPs felt unwilling to challenge the GPs’ antimicrobial prescribing. This attitudinal issue of “not to challenge or be challenged” might hinder optimal communication between GPs and GPs on matters related to AMS.

#### 2.3.4. Ignorance of Patient’s Clinical Conditions by the Pharmacists

Some GPs were concerned about the lack of clinical training of CPs about family care and lack of understanding of patient’s clinical conditions that may lead to inappropriate recommendations on antimicrobial therapy. On the other hand, CPs felt that if they had better access to the patient’s clinical records, then they would feel more competent about making recommendations. Both GPs and CPs reported that the current form of ‘MHR’ was of little help and was not easy to use. Many CPs recommended that integration of clinical indications for the prescribed antimicrobials within the ‘MHR’ would facilitate their interventions.

Some GPs agreed that the clinical decisions on the choice of antimicrobial(s) prescribed should be based on educational advice, following guidelines, and communication with pharmacists. However, CPs felt difficulty in judging adherence to guideline because they are often unsure about the infection being treated or the clinical indications for which antimicrobial(s) have been prescribed. Both GPs and CPs were cognisant of the boundary of their roles in AMS and that inter-professional communication should not be about judging the individual’s professional competencies.

#### 2.3.5. Experience of the GP

Some GPs thought that collaboration would be more appropriate for newly qualified doctors than experienced GPs. They disagreed that a CP should influence prescribing of experienced doctors but felt that the advice from CPs would benefit junior doctors. A similar concern was expressed by CPs that it would be difficult to convince more experienced GPs to listen to them and accept any recommendations for changing antimicrobial therapy.

#### 2.3.6. Patient Related Factors

Some GPs reported that not all patients want the pharmacists to know their diagnosis. Consistently, CPs mentioned that it is difficult to convince a patient after a doctor has provided a prescription for antimicrobials, and that patients also do not like CPs to contact the patients’ GPs to clarify on the appropriateness of the prescription. Few CPs reported that when they explained to customers that an antibiotic was unnecessary, the customer responded that they preferred to do what their doctor has told them. Therefore, the patient’s awareness about the extended role of CPs beyond just dispensing is critically important. Furthermore, the patient’s acceptance of collaborative antimicrobial care is equally important to develop a GP–CP collaborative model of care for AMS in primary care.

#### 2.3.7. AMS Supportive Tools and Technologies

Both GPs and CPs highlighted potential problems in the availability of required tools and technology such as a clinical decision support system, improved version of prescribing and dispensing software that is integrated with clinical guidelines or prompts, communication technology (e.g., telehealth), a patient’s diagnostic reports (e.g., antibiogram report), and point-of-care tests facilities to avoid diagnostic uncertainty. Both GPs and CPs believed that a linking software that integrates AMS resources with the prescribing and dispensing software, communication tools that guide how to communicate inter-professionally and with the patients, and telehealth technologies that collectively may foster triage services, are important to facilitate optimal antimicrobial therapy. CPs recognised the paramount importance of having a usable form of MHR that would enable them to scrutiny a patient’s clinical and medicine information while making a recommendation on the choice of antimicrobial and interact with GPs where required.

#### 2.3.8. Communication and Access

Both GPs and CPs reported limited opportunities for inter-professional communication with regards to antimicrobial prescribed. Both reported difficulty in setting up communications when patients were with the GPs or CPs. According to the participating CPs, there was no existing collaborative care approach or formal patient referral system that supported good communication with GPs regardless of patient’s type of infection and class of antimicrobials prescribed by GPs. Time constrains and lack of timely access to each other is a major challenge for good communication.

## 3. Discussion

This is the first primary care study to explore interprofessional attitudes and challenges in implementing GPPAS strategies from the perspective of both GPs and CPs, a critical but poorly delineated dimension. Most GPs and CPs had a good understanding of the objectives and impact of AMS programs although nearly 30% still did not perceive that their individual effort would make a difference in AMR and AMS. This misperception is lower among study CPs compared to the median value of 51.4% derived from 10 CP-AMS surveys around the world [20]. The majority of participating GPs and CPs agreed that they would require adequate training to conduct AMS; interprofessional activities and checklists for AMS were also not in place to collaboratively work with antimicrobial prescriptions. Both groups were convergent and supportive of GP–pharmacy practice agreements and the policies that would enhance a GP–CP collaboration for improving AMS practice. GP–CP collaborative AMS models facilitating interprofessional group meetings and antimicrobial audits for quality improvement of antimicrobial prescribing were more strongly supported by CPs compared to GPs. Nevertheless, GPs were less confident and had divergent views about the prospective benefits of pharmacist’s co-location in general practice and the use of MHR, respectively, in optimising antimicrobial therapy and improving GP–CP communications with antimicrobial prescriptions. Overall, there is an opportunity to implement AMS by an increased GP–CP collaboration but a range of addressable barriers and challenges were identified.

Our findings suggest that the current GP–CP collaboration in AMS is a piecemeal process, but increased trust towards each other’s professional competency and appreciation of each other’s AMS roles might foster good collaboration. The lack of interprofessional trust on, and training to improve, professional AMS competency was consistent with the published literature [20,25]. In the UK, a national consensus-led competency framework has been used to provide a basic set of AMS competencies among undergraduate healthcare professionals including GPs and CPs [26]. The importance of interprofessional trust was highlighted by Australian CPs when they collaborate with GPs under the home medicine review (HMR) program [27]. Van et al. [28,29] also showed that GP–CP collaborative behaviours are directly influenced by interactional determinants (e.g., communication, mutual respect, and role recognition) and indirectly influenced by individual determinants (e.g., trust and expectation).

Thus, for AMS to occur, at an individual level, GPs and CPs need to be open enough to receiving or providing intervention and accepting feedback about an antimicrobial prescription. Study GPs were comparatively (*p* < 0.001) less supportive to GP–pharmacist collaborative antimicrobial audit (46.1% vs. 86.5%) model than CPs. This attitude of GPs emphasise that the feedback on antimicrobial prescription should not be seen as a criticism but a pathway for quality improvement and patient safety. In this regard, the motivation of the participating GPs and CPs to the co-construction of AMS knowledge is critically important; co-construction might occur by sharing areas of problematic antimicrobial prescribing decisions, exploring dissonance in opinion, and consensus development about a safer antimicrobial therapy between themselves. The application of the concept of co-construction of knowledge [30] might have a greater importance to develop an interprofessional AMS learning process. In this regard, as site champions, a local GP–CP interprofessional team can be developed to promote the uptake of AMS resources and audit the compliance of AMS practice in Australia. Such a model in Scotland was effective to reduce the use of broad-spectrum antimicrobials in a large region as part of a national initiative [17].

Though majority of our participating GPs and CPs were interested to participate in the regular GP–CP antimicrobial pharmacotherapy group meetings, an attitudinal divergence was significant (GPs vs. CPs; 54.9% vs. 82.5%; *p* < 0.001). However, in our qualitative findings, establishing a GP–CP practice agreement was viewed as important by both to facilitate this group meetings model. A randomised controlled trial in Netherland demonstrated the improvement of antibiotic prescribing through a system supported GP–CP collaborative pharmacotherapy audit meetings [31]. According to a 2020 study, multimodal AMS education by a primary care team including GPs and pharmacists had a long-term impact on sustained reduction of antibiotic prescribing and infections caused by *E. coli* in the community [32].

According to our study CPs, patients have a lack of awareness about CPs’ extended roles/services including communication with GPs to optimise antimicrobial therapy. According to a UK study [33], the awareness and attitudes of patients were influenced by how frequently patients used the services (such as sore throat test by CPs) and felt benefited from using the services. Patients’ use of, and access to, new pharmacy services (e.g., point-of-care tests followed by appropriate referral to GPs) depends on the professionalism of the team of CPs and how strong the GP–CP collaboration is as well [34]. Hence, limited CP-training and access to using patient information leaflets or point-of-care tests in Australian community pharmacies might partly explain the patients’ gap in awareness about GP–CP collaborative antimicrobial care.

Less than a quarter of our participants with no significant difference between GPs and CPs (*p* < 0.116) used the patient information leaflets while treating patients with infections, reflecting a limited provision of using patient-forwarding antibiotic information leaflets. In England, a cluster RCT [35] with 272 community pharmacies demonstrated that the use of leaflets (self-care guides) for patients with respiratory infections (RTIs) was significantly associated with lower referral to GP for certain RTIs patients and CPs were able to manage self-limiting infections. This effective intervention would be worthy of investigation in Australia.

Similarly, the uptake of point-of-care tests by GPs and CPs was below 20% with no interprofessional difference (*p* < 0.784). In the US settings, point-of-care tests service in community pharmacies to optimise antimicrobial use in pharyngitis and influenza management was effective, feasible and acceptable to patients when a local GP–pharmacy collaborative practice models supported the patient referral system [18,36]. This indicates that in the study context, improved patient referral system, collaborative practice agreement, and professional and diagnostic AMS training of CPs would be important to introduce a new GP–CP collaborative diagnostic AMS model. More importantly, these improvements in future would increase two-way patient referrals between GPs and CPs while using point-of-care tests to make sure either patients need antibiotics or OTC or visit to GPs/CPs to optimally treat the infections.

Our study emphasises that some key structural developments are required to improve GP–CP interactions about antimicrobial prescriptions and optimally manage primary care patients with infections. The current form of “MHR” did not allow CPs to communicate with GPs regarding antimicrobial prescriptions in many cases where indications or reasons for prescribing antimicrobials were missing. This prevents CPs to comment on GPs’ antimicrobial prescribing. Additionally, a lack of IT support, timely access to diagnostic and antibiogram reports, and availability to each other that hindered study CPs’ ability to query or communicate with GPs. Jeffs et al. [37] echoed these barriers what was commonly faced by primary care prescribers including pharmacists when making decisions about antimicrobial(s). The mandatory integration of clinical indications into patients’ MHRs and telehealth-led reviewing of antimicrobial prescription(s) might increase the digital interpretability of antimicrobial prescriptions and case-conferencing where necessary to assure appropriateness of antimicrobial recommendations by CPs. Importantly, these fundamental improvements require greater attention and commitment by the AMS stakeholders and funders.

The use of the Therapeutic Guidelines: Antibiotics [24] was not a common practice among CPs; significantly lower than GPs (CP vs. GP; 45.5% vs. 83.2%; *p* < 0.0001). This might lead a lack of confidence of CPs in how to use these guidelines for assessing appropriateness and guideline-adherence of antimicrobial prescription(s). Furthermore, study GPs and CPs indicated that they were not regularly updated about the AMR reports (local, regional or community level) that might help to predict the effectiveness of antimicrobials prescribed to the patients. These are important points to consider when designing AMS training modules or resources or local antimicrobial guideline specific for GPs and CPs. The increased confidence of CPs in using guideline and their AMR updates might facilitate interprofessional communications with antimicrobial prescription(s).

Antimicrobial audits and providing feedback for quality improvement is a proven, effective, and sustainable model where GPs and CPs need greater involvement to improve the quality of antimicrobial use. Interestingly, our study participants demonstrated positive attitudes towards the interprofessional involvement in antimicrobial audits. However, the quality indicators for the use of antimicrobials, electronic database that ease clinical audits and data extraction, a dedicated interprofessional team and the context-specific tools to audit antimicrobial prescriptions are lacking in Australian community. If these resources were available, CPs could be more engaged in performing antimicrobial audits, reviewing antimicrobial prescriptions, and providing feedback to GPs. In Australia, a tool [38] to audit antimicrobial use in the context of remote primary healthcare has been tested and this tool could be integrated into clinical practice in future.

In the current study, though most CPs were highly supportive of their interprofessional role in AMS being defined by their professional organisations, GPs were hesitant. According to a Cochrane review [39], use of interprofessional checklists can improve the compliance to recommended stewardship practice. To date, there are many misunderstandings and misperceptions in relation to AMR and AMS exist with the tools used to facilitate interprofessional communication, communication with the patients, and the standard care [40]. This is where GP and pharmacy professional organisations in Australia should come forward with a research agenda of how to develop a clear consensus guidelines defining interprofessional activities including checklists to improve the routine AMS practice by GPs and CPs. This leadership role of professional organisations might be vital to create a culture where both GPs and CPs would feel professionally supportive to take stewardship responsibility in optimising antimicrobial use in primary care.

In summary, our results tell us that Australian CPs are significantly more likely than GPs to feel that (a) better collaboration is needed; (b) regular GP–CP group meetings are needed; (c) GPs should be open to advice from CPs about the selection of antimicrobial agent and dose; (d) the placement of a CP in general practice; (e) use of point-of-care tests using GP-community pharmacy collaborative practice agreement; (f) electronic exchange of antimicrobial prescription and shared electronic patient records are required for CP-led comprehensive reviewing. This attitudinal divergence still remains as a greater impediment to effective GP–CP collaboration in AMS.

The CPs in a few countries such as the UK, USA and the Netherlands are further ahead in their engagement in promoting AMS programs [19,41,42] compared to Australia perhaps due to health system being more supportive to and positive attitudes of GPs and CPs towards GP–CP collaboration. We assume the development of a feasible GP–CP collaborative AMS model would help use the expertise of Australian CPs in promoting AMS, assist them to develop AMS competency, and collectively, GPs and CPs could contribute in reducing AMR in primary care. Further research is needed to test the validity of the attitudinal model of GP–CP collaboration in AMS for the identified determinants influencing GP–CP collaboration. Patients’ views about the awareness and acceptability of collaborative services for optimal antimicrobial care should not be overlooked. Given the multimodal complexities in the system and future opportunities for GP-CP collaboration, a multi-stakeholder approach using the concept of co-design and co-production can be undertaken to build a feasible GPPAS model to improve AMS in primary care.

### 3.1. Strengths

To our knowledge, this is the first study internationally that compares and contrasts the views of GPs and CPs about the collaborative implementation of AMS, both quantitatively and qualitatively, through two nationwide surveys. In the hospital settings, there is much evidence for strong physician-pharmacist collaboration for effective implementation of AMS programs [38,39]. In contrast, this study is very important because the hospital collaborative model is not feasible in the primary care setting, and the results will assist the design of such a model in primary care. Though response rates of our surveys seem low, the response rates of this study were greater than the average response rates of Australian GPs and CPs to surveys [43,44,45,46]. Our survey tools demonstrated good reliability (Cronbach’s alpha >0.8). Our results are based on 999 participants including 386 GPs and 613 CPs nationally, with response rates of both GPs and CPs proportionate to the Australian national workforce statistics and thus comparable. We reached a minimum sample size of 336 (112 in one group and 224 in another group) that was required to detect a difference between two proportions (GPs vs. CPs) of at least 15 for an α level, 0.05 and β level, 0.20 with affording 80% power.

### 3.2. Limitations

Response biases might be present due to the voluntary nature of participation in surveys and for CPs to respond both online and using paper-based questionnaire. The results of free-text analyses are not generalisable though qualitative results were derived from 813 survey participants. The open-ended questions of the surveys were not semi-structured. However, these questions gave participants the freedom to express exactly how they feel about the practical difficulty of implementing AMS without limiting them to some topic, and it allowed expression of unique ideas that may reveal unforeseen challenges and opportunities of GP–CP collaboration in AMS.

## 4. Materials and Methods

This study was a part of an overarching research project to develop a GPPAS model in improving AMS implementation in Australian primary care. We framed this study comparing two nationwide surveys of GPs and CPs on AMS to explore interprofessional perspectives of implementing AMS in collaboration between GPs and CPs. We undertook two nationwide surveys of GPs and CPs across Australia between January and October in 2019.The detailed methodologies of the survey design and survey tools used are found in the published literature [47,48]. The two-stage stratified random sampling method was employed to select nationally representative samples of 2500 GPs and 2160 CPs using the Australasian Medical Publishing Company (Ampco) database [49] and a pharmacy database. Two survey tools were developed involving the representatives from the National Centre for Antimicrobial Stewardship (NCAS) and the Department of General Practice of Monash University. A systematic review of GP–pharmacist collaborative AMS strategies, a scoping review of 10 AMS surveys of CPs, and other AMS surveys of GPs were the sources of the survey items [16,20,47].

Twenty quantitative items and two qualitative items were common among two survey tools used for GPs and CPs. Of the 20 common survey items, 5 items were related to perceptions of AMS programs, four items were related to practices of using AMS strategies during patient consultation, seven items were related to attitudes towards GP–CP collaborative AMS approaches, and five items were related to the attitudes towards strategies for future improvement of AMS including GP–CP collaboration. A 5-points Likert scales of ‘strongly agree to strongly disagree’ and ‘always to never’ were used. The two qualitative questions explored perceived barriers to, and facilitators for, implementing AMS in a collaborative manner. Participants responded to the open-ended questions in the form of free text.

Upon Ethics approval by the Monash University Human Research Ethics Committee (MUHREC), we successfully invited 2500 GPs and 2000 CPs by mail. Participants were sent a survey-package which had an invitation letter explaining the research, a paper-based survey questionnaire, and a reply-paid envelope. CPs were additionally provided access to an online survey link on the invitation letter and asked to participate either online or by using the paper-based questionnaire provided.

For reporting of the survey results, a published guideline was used [50]. Descriptive statistics were used to determine the frequency of distribution and median of survey responses. The agreement responses of ‘strongly agree to strongly disagree’ was grouped into agree, neutral and disagree and ‘always to never’ into always/often, occasionally, and rare/never. Comparing proportion-Chi square tests was applied to determine the agreement response variation between GPs and CPs. We compared responses of 20 items individually to identify the differences and similarities in perceptions regarding awareness of AMS programs, use of AMS strategies, attitudes towards inter-professional collaboration in AMS and future improvement strategies. We used Microsoft Excel (Microsoft Office 365) and SPSS version 24.0 for analysing the quantitative data.

The qualitative responses in the form of free-text were extracted and imported into Monash University Microsoft office 365 excel file. The Systems Engineering Initiative for Patient Safety (SEIPS) 2.0 model [51] was used as a theoretical framework to guide the processing and analyses of qualitative data. The SEIPS model explains how a work system (e.g., general practice, community pharmacy) influence a process (AMS practices) to impact in patient safety (safe antimicrobial use). All written texts were read by two researchers (SKS and SP) independently and categorised into model components consisting of person factors, tools, and technology factors, organisational factors, environmental factors and others separately for both GPs and CPs. We calculated the percentage of GPs and CPs whose comments were fell into SEIPS model components (themes) and subthemes using the concept of content analysis method [52]. Then data were dual coded to identify the common subthemes and discrepancies were resolved by discussion meetings. The more common subthemes under each broad theme were selected based on the prevalence measures (percentage of GPs and CPs fell into each subthemes) to avoid selection bias. We then identified and compared the major themes, subthemes, and representative quotes (Table 3) under the domains of SEIPS model using the principles of qualitative data analyses-thoughtful interrogation and examination of the data, critical reflection, and insightful reporting [53].

## 5. Conclusions

There are opportunities for GP–CP collaboration to foster AMS implementation in Australian primary care. However, significant challenges remain in how to improve interprofessional education, trust, and competencies for AMS. There is a need for GPs and CPs to recognise their interprofessional roles in identifying patients who truly need antimicrobials, and improving the choice, dose, and duration of antimicrobials to reduce avoidable AMR. The arrangement of health system structure and policies that improve the GP–pharmacy collaborative practice agreements would support the development of a GP–CP collaborative antimicrobial stewardship model so called GPPAS in Australia.

## Figures and Tables

**Table 1 antibiotics-10-00047-t001:** Demographics of survey participants responded to quantitative and open-ended questions.

Demographics	General Practitioners (GPs) n (%)		Community Pharmacists (CPs) n (%)	
Total Responses	Quantitative Responses (N = 386)	Qualitative Responses (N = 212)	Quantitative Responses (N = 613)	Qualitative Responses (N = 592)
Sex				
Male	195 (51.1%)	112 (50.6%)	272 (44.4%)	268 (45.3%)
Female	186 (48.8%)	109 (49.4%)	341 (55.6%)	324 (54.7%)
Education				
B. Med science/B. Pharm	4 (1.0%)	4 (1.8%)	518 (84.6%)	515 (87.0%)
MBBS	305 (79.4%)	150 (67.8%)	-	-
MD	31(8.0%)	28 (12.6%)	-	-
Masters	39 (10.1%)	35 (16.8%)	71 (11.6%)	71 (12.0%)
PhD	5 (1.30%)	4 (1.8%)	6 (1.0%)	6 (1.0%)
Years of GP/pharmacy practice				
≤5	20 (5.2%)	18 (8.1%)	159 (26.1%)	156 (26.4%)
610	43 (11.1%)	40 (18.0%)	142 (23.3%)	139 (23.5%)
>10	322 (83.6%)	163 (73.7%)	308 (50.6%)	297 (50.2%)
Current GP practice/pharmacy practice location				
Metro	234 (60.9%)	97(43.9%)	328 (53.9%)	322 (54.4%)
Regional	74 (19.2%)	58 (26.2%)	146 (24.0%)	140 (23.6%)
Rural	62 (16.1%)	54 (24.4%)	122 (20.1%)	119 (20.1%)
Remote	14 (3.6%)	12 (5.4%)	12 (2.0%)	11 (1.9%)
State of work				
NSW (New South Wales)	104 (27.0%)	60 (27.1%)	137 (22.4%)	134 (22.6%)
VIC (Victoria)	105 (27.2%)	45 (20.3%)	105 (17.0%)	101 (17.1%)
QLD (Queensland)	73 (18.9%)	42 (19.0%)	139 (22.7%)	134 (22.6%)
ACT (Australian Capital Territory)	5 (1.2%)	4 (1.8%)	9 (1.5%)	9 (1.5%)
SA (South Australia)	39(10.1%)	22 (9.9%)	95 (15.5%)	94 (15.9%)
WA (Western Australia)	36 (9.3%)	18 (8.1%)	72 (11.8%)	71 (12.0%)
TAS (Tasmania)	18 (4.6%)	15 (6.7%)	47 (7.7%)	41 (6.9%)
NT (Northern Territory)	5 (1.3%)	5 (2.2%)	9 (1.5%)	9 (1.5%)
Medical/Pharmacy training				
Outside Australia	124 (32.2%)	70 (31.6%)	61 (10.0%)	48 (8.1%)
Inside Australia	261 (67.7%)	151(68.3%)	552 (90.0%)	544 (91.9%)
Completion of antimicrobial modules				
Yes	105 (27.4%)	86 (38.9%)	115 (18.8%)	112 (18.9%)
No	200 (52.2%)	103 (46.6%)	280 (45.8%)	270 (45.6%)
Not aware	78 (20.3%)	32 (14.4%)	216 (35.4%)	209 (35.3%)

**Table 2 antibiotics-10-00047-t002:** Convergence and divergence in the attitudes of GPs and CPs about AMS programs and collaborative implementation.

Survey Items	CP’s Agreement		GP’s Agreement		*p*-Value	95% CI
**AMS programs**						
I am familiar with the term antimicrobial stewardship (AMS)	72.9	447/613	68.9	266/386	0.1735	−1.72% to 9.85%
AMS programs in my practice will significantly reduce inappropriate use of antimicrobials	66.8	409/612	61.7	237/384	0.1010	−0.97% to 11.23%
AMS programs will reduce health care costs associated with infections	83	508/612	70.8	273/383	< 0.0001	6.82% to 17.69%
Individual efforts at AMS have minimal impact on the problem of antimicrobial resistance	32.7	200/612	24.6	204/383	<0.0065	2.29% to 13.66%
I require adequate training to undertake AMS	76.5	468/612	46.4	179/385	<0.0001	23.99% to 35.96%
**Use of AMS strategies**						
I use national antimicrobial guidelines when prescribing/dispensing antimicrobials to my patients	45.5	274/602	83.2	321/385	<0.0001	32.00% to 42.90%
I educate my patients or their carers about unintended consequences of antimicrobial use like antimicrobial resistance, impact on gut microbiota etc.	76.8	467/608	82.4	316/383	0.0353	0.38% to 10.55%
I share patient information leaflets about infections when I counsel my patients or carers who require antimicrobials or may have an infection	24.5	149/608	20.2	78/384	0.1162	−1.09% to 9.45%
I use rapid point-of-care tests to guide my clinical decision about whether to prescribe/dispense an antibiotic to the patients with pharyngitis or the flu	19.1	114/596	18.4	71/382	0.7848	−4.42% to 5.59%
**Attitudes towards GP–pharmacist collaboration**						
Improving AMS in the community will need a policy that supports better collaboration between general practice and pharmacy	92.4	560/606	60.9	235/381	<0.0001	26.16% to 36.81%
Pharmacists with knowledge of antimicrobials and infections should attend regular group meetings of GPs within general practice clinic to discuss antimicrobial pharmacotherapy	82.5	509/605	54.9	212/381	<0.0001	21.71% to 33.35%
GPs should be receptive to pharmacists providing advice about the choice of antimicrobial prescribed	92.6	561/606	63	195/382	<0.0001	24.34% to 34.87%
GPs should be receptive to pharmacists making recommendations in consultation to the doses or formulations of the antimicrobial prescribed	93.6	567/606	50.5	244/381	<0.0001	37.63% to 48.37%
A pharmacist co-located within general practice can help optimise antimicrobial therapy of patients with infections	79.5	482/606	39.8	154/382	<0.0001	33.66% to 45.35%
An electronic prescription exchange technology between GP and pharmacy should be introduced for reviewing the appropriateness of antimicrobial prescriptions	74.3	449/605	26.3	140/382	<0.0001	42.11% to 53.32%
The “My Health Record” could improve communication between GPs and community pharmacists about antimicrobial prescriptions	67.2	416/604	30.9	119/382	<0.0001	30.14% to 42.01%
**Future needs to practice AMS**						
I would be willing to participate in a program of training focused on AMS	87.3	529/606	72	278/386	<0.0001	10.16% to 20.56%
I support the introduction of standard guidelines to assist in the implementation of AMS programs	93.6	566/605	80	309/386	<0.0001	9.28% to 18.19%
I support a policy that limits accessibility of some antimicrobials in the community	69.5	420/604	74.4	287/386	0.0962	−0.88% to 10.47%
Professional GP/pharmacy organisations should define my roles and responsibilities regarding AMS activities	74.6	449/602	39.9	153/382	<0.0001	28.53% to 40.52%
I support the involvement of a specialist physician and a pharmacist providing individualised antimicrobial prescribing advice and feedback to GPs	86.5	523/604	46.1	178/386	<0.0001	34.60% to 45.90%

**Table 3 antibiotics-10-00047-t003:** Barriers and challenges of AMS implementation by GP–pharmacist collaboration.

Domains and Themes	GPs’ perspective (Representative Quotes)	CPs’ perspectives (Representative Quotes)
**Person level**	**GPs (44.8%, 99/221)**	**CPs (56.4%, 334/592)**
**Professional AMS Training** **(GP vs. CP; 80.8% vs. 94%)**	*There is non-awareness of practitioners of the impact of use of antibiotics to resistance mechanisms and long and short- term effects to patients (GP-280)* *Poor undergraduate and post graduate training in antimicrobial pharmacology (GP-1547)*	*Lack of specific knowledge about suitability of specific antibiotics for specific infections (CP-141)* *Lack of pharmacists training in professional courses relevant with the AMS (CP-91)*
**Interprofessional Trust** **(GP vs. CP; 23.2% vs. 28.4%)**	*I am not sure that community pharmacist the best person to educate me about AMS. Some doctors find it offensive depending on the approach (GP-1908)* *Happy to take advice. Not happy to be constrained by someone who does not know the patient and their situation as well as I do-it needs to be collaborative and pharmacist should acknowledge my expertise and knowledge of patients/family/background circumstances (GP-1878)*	*Doctors don’t believe that pharmacists have the knowledge to advise them on antimicrobial. I’m a graduate of master of infectious diseases from UWA and master of public health from Newcastle university with 8 years of experience working as an antimicrobial stewardship and ID pharmacist. When I discussed case to the doctors, they always just say they think O/C to go ahead (CP-426)* *GPs have lack of confidence in pharmacist’s knowledge of antimicrobials therapy (CP-139)*
**Resistance** **(GP vs. CP; 36.4% vs. 22.2%)**	*I am very unwilling to have the pharmacist telling me what to do (GP-765)*	*GPs not always receptive to my recommendation or queries regarding antibiotic choice/ dose/regimen (CP-285)* *When approach, some GPs do not consider my advice. One GP in particular routinely prescribes Clarithromycin 500mg BD to any patient with a cough. I have advised this may be an unnecessary high dose to no effect. I feel quite powerless in this situation (CP-303)*
**Unwillingness to challenge or be challenged** **(GP vs. CP; 28.3% vs. 15.6%)**	*GPs might not to be receptive of pharmacists interfering and advising them after their consent (GP-1470)*	*It’s difficult when the GP has written a script. Therefore, it feels like undermining them (CP-448)* *Some GPs are very hard to collaborate with especially when they feel threatened by their decision being challenged (CP-327)* *GP’s have feelings that we are overstepping their justification (CP-401)*
**Experience of GPs** **(GP vs. CP; 14.1% vs. 19.6%)**	*Collaboration would be more appropriate for newly qualified doctors rather than experienced GPs. I disagree that a chemist should influence on experienced doctors prescribing but can see a benefit for newly qualified doctors (GP-2544)*	*Older GPs are not very accepting of advice unless it is along the level of alternatives if something is T.0.0.S (out of stock) (CP-388)* *GPs accept advice from pharmacists. Your biggest hurdle will be to convince more senior GPs to listen to pharmacists and accept any AMS guidelines (CP-199)*
**Patients** **(GP vs. CP; 48.5% vs. 50.9%)**	*Not all patients want the pharmacist to know their diagnosis (GP-2633)*	*When I explain to customers that an antibiotic maybe unnecessary, I am met with comments such as ‘I prefer to do what the doctor said’(CP-303)* *Patient don’t like us to contact a GP for appropriateness of what GP prescribed (CP-140)*
**Ignorance of patient’s clinical conditions** **(GP vs. CP; 42.4% vs. 23.4%)**	*Pharmacist have little or no clinical training and are not involved in total patient family care. Getting objective advice with little knowledge of the full circumstances can cause problems esp. if patients are counselled by pharmacists without first approaching the GPs (GP-2934)* *How can a pharmacist provide advice when they do not know the clinical indication for a script? (GP-1329)*	*We don’t have the indications of antibiotic prescriptions, so guideline checking difficult (CP-458)*
**Concern of assessing guideline- adherence** **(GP vs. CP; 12.1% vs. 19.5%)**	*The clinical decisions should be based on educational advice and following guidelines, correctly pharmacist can advise on doses but not choice (GP-2262)*	*Can’t check appropriateness against guidelines and unsure what infection is being treated (CP-482)*
**Conflict of interest** **(GP vs. CP; 24.2% vs. 0%)**	*Pharmacist have a conflict of interest, they get paid for prescribing something and may make more profit from OTC cough/cold preps (GP-83)* *Pharmacist so often give not only advise which is wrong but even change the generics (GP-12)*	*-*
**Tools and Technology level**	**GPs (29.9%, 66/221)**	**CPs (26.4%, 156/592)**
**Logistics** **(GP vs. CP; 80.3% vs. 84.6%)**	*Therapeutic guidelines should be integrated with the prescribing software (GP-548)* *My health record in its current form is of minimal help for anything (GP-854)*	*Fully understandable clear guidelines for pharmacists doing AMS are not readily available (CP-1)* *References including Therapeutic Guidelines and product information are not always clear on the duration of an antibiotic course (CP-498)* *AMS resources linked to dispensing program rather than looking up texts (CP-133)* *Medical Health Record is not easy to use (CP-386)*
**Interaction level**	**GPs (29.4%, 65/221)**	**CPs (34.5%, 204/592)**
**Communication** **(GP vs. CP; 61.5% vs. 68.8%)**	*I have a lack of communication with pharmacist… (GP-1021)*	*Pharmacist has limited and untruly access to GP-many will nor call not get the messages (CP-559)* *Difficulty in communication with GPs in a timely manner (CP-63)*
**Group learnings/meetings** **(GP vs. CP; 36.9% vs. 29.7%)**	*GPs work independently. No practice meetings occur regarding strategy of optimal antimicrobial prescribing (GP-608)*	*GP–pharmacist group meetings may be hard to arrange in rural pharmacy depending on where it is organised due to restriction on travel/distance (CP-222)* *The real problem for meetings with doctors is which pharmacists go to which doctors’ meetings? (CP-190)*
**Collaborative health system structure** **(GP vs. CP; 0% vs. 25%)**	*-*	*No collaborative care or system between pharmacists and GPs regardless of type of infection-pharmacists rely on the patient for information regarding infection (CP-478)* *Ability to contact GPs instantly for referral, not readily available (CP-343)*
**Staff shortage** **(GP vs. CP; 0% vs. 35.2%)**	*-*	*Lack of pharmacist make it difficult to check every single script against AMS guidelines and spend enough time with every patient to make sure they fully understand the issues (CP-237)* *Staffing, Pharmacy board says up to 150 scripts/day is acceptable for single pharmacist and in unforeseen circumstances can go higher! do the math 7-patient expectation (fast dispense), most successful pharmacies are bulk discount models, why to invest in the unknown, lack of support from infectious disease specialist or at least GP with interest 11-in regional and remote we heavily rely on locums and you can’t be very picky! (CP-528)*
**Environment level**	**GPs (66.1%, 146/221)**	**CPs (39.9%, 236/592)**
**Time** **(GP vs. CP; 87% vs. 60.6%)**	*Busy GP environment and a lack of time and to determine and explain appropriate antibiotics to patients and also to discuss with pharmacists (GP-2835)* *I have to use my valuable time if direct communication with pharmacists is required (GP-1187)*	*We have an almost impossible task even getting through to speak to local GPs on matters of drug interactions and often it takes 48 to 72 hours before they respond to a phone call (CP-431)* *Difficulty to speak to GP while the patient is waiting in the pharmacy (CP-557)*
**Access to diagnostic reports** **(GP vs. CP; 17.3% vs. 38.6%)**	*Antibiogram reports are not readily accessible (GP-562)*	*Lack of insight in most occasion into indications/lab testing/culture results. Therefore, affecting our ability to assess suitability of antimicrobial prescription (CP-360)* *Pharmacist relies on the patient’s willingness to discuss the nature of their infection-if the antibiotic is broad spectrum. At present the pharmacist relies on the patient’s interpretation of the diagnosis. Even not all patients allow access to pharmacists for e-health record (CP-423)*

## Data Availability

All data related to this study are available and accessible on request from the corresponding author.

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
