# Peer review of "Divergent and Convergent Attitudes and Views of General Practitioners and Community Pharmacists to Collaboratively Implement Antimicrobial Stewardship Programs in Australia: A Nationwide Study"

_antibiotics, 2021, doi:10.3390/antibiotics10010047_

Round 1

Reviewer 1 Report

This is an excellent manuscript addressing the major challenges to the successful implementation of effective community-based antimicrobial stewardship programs, namely the gulf that continues to exist between community-based clinical pharmacists and general practitioners. The overall design of the study and manuscript are strong and the conclusions are supported by the data, but I have two reservations:

  1. It was not clearly stated in the methods section how the highlighted comments in Table 3 were chosen (the last sentence under (2.3) states "relevant quotes"), which leads to the potential for selection bias. I like that you have touched on this in describing study limitations. You describe a detailed (SEIPS) method utilizing two independent researchers to read and categorize the comments, but was there a method to determine what percentage of GPs and CPs fell into each category? I feel that the paper would be strengthened if the comments in Table 3 could include such a measure, or if you could choose the single quote representative of the highest percentage of GPs and CPs in each category.
  2. In my interpretation the results tell us that a large gulf continues to exist between GPs and CPs with respect to attitudes toward collaboration. The first sentence in (2.2.3) does not reflect this. The results tell us that CPs are significantly more likely than GPs to feel that (a) better collaboration is needed; (b) regular meetings are needed; GPs should be open to advice from CPs about (c) selection of agent and (d) dose; (e) the placement of a CP in general practice; (f) electronic prescription exchange; and (g) shared electronic patient records. This remains a huge impediment to effective collaboration, and is rightly the major focus of your discussion. However, even in the discussion the magnitude of this difference is not clearly shown. For example, paragraph 4 of the discussion begins with "Majority of our participating GPs and CPs were interested to participate...." Strictly speaking yes, but it is 82.5% of CPs compared to only 54.9% of GPs (P<0.001); this is a major difference of opinion, which should be emphasized in the discussion. You give very good examples from multiple countries where collaborative efforts have been implemented; I would also suggest addressing the potential impact of inter-professional educational activities on future collaborative efforts. Medical, pharmacy, and nursing schools currently integrate a number of shared case-based learning activities into their respective curricula, beginning early (eg., first-year students) and continuing throughout the respective curricula.

Author Response

This is an excellent manuscript addressing the major challenges to the successful implementation of effective community-based antimicrobial stewardship programs, namely the gulf that continues to exist between community-based clinical pharmacists and general practitioners. The overall design of the study and manuscript are strong and the conclusions are supported by the data, but I have two reservations:

Thank you for your comments highlighting the strength of the manuscript and indicating two areas where there is a scope for further strengthening of the manuscript. All amendments and additions in responses to your comments have been done using track changes.

1. It was not clearly stated in the methods section how the highlighted comments in Table 3 were chosen (the last sentence under (2.3) states "relevant quotes"), which leads to the potential for selection bias. I like that you have touched on this in describing study limitations. You describe a detailed (SEIPS) method utilizing two independent researchers to read and categorize the comments, but was there a method to determine what percentage of GPs and CPs fell into each category? I feel that the paper would be strengthened if the comments in Table 3 could include such a measure, or if you could choose the single quote representative of the highest percentage of GPs and CPs in each category.

Now, we have detailed in the method section about how subthemes and quotes were chosen. We calculated the percentage of GPs and CPs whose comments were fell into SEIPS model components (themes) and subthemes using the concept of content analysis method. This helped to avoid selection bias. The reference 52 describing the method has been added in the reference list.

We have amended table 3 putting the percentage measures of GPs and CPs whose comments feel into each domain of SEIPS model, and also for each subtheme.

We have also reduced a number of quotes from table 3 and kept two representative quotes under each subtheme for better understanding of the readers.  

2. In my interpretation the results tell us that a large gulf continues to exist between GPs and CPs with respect to attitudes toward collaboration. The first sentence in (2.2.3) does not reflect this. The results tell us that CPs are significantly more likely than GPs to feel that (a) better collaboration is needed; (b) regular meetings are needed; GPs should be open to advice from CPs about (c) selection of agent and (d) dose; (e) the placement of a CP in general practice; (f) electronic prescription exchange; and (g) shared electronic patient records. This remains a huge impediment to effective collaboration, and is rightly the major focus of your discussion. However, even in the discussion the magnitude of this difference is not clearly shown. For example, paragraph 4 of the discussion begins with "Majority of our participating GPs and CPs were interested to participate...." Strictly speaking yes, but it is 82.5% of CPs compared to only 54.9% of GPs (P<0.001); this is a major difference of opinion, which should be emphasized in the discussion. You give very good examples from multiple countries where collaborative efforts have been implemented; I would also suggest addressing the potential impact of inter-professional educational activities on future collaborative efforts. Medical, pharmacy, and nursing schools currently integrate a number of shared case-based learning activities into their respective curricula, beginning early (eg.,first-year students) and continuing throughout the respective curricula.

Thank you for rightly interpret the results. We have included a likely summary paragraph at the end part of the discussion. The first sentence in (2.2.3) has been amended. In the discussion section, the magnitude of this difference has been given in all relevant places for more clarity; amended sentences are as follows.

“Study GPs were significantly (p<0.001) less supportive to GP-pharmacist collaborative antimicrobial audit (46.1% vs 86.5%) model than CPs.”

“Though majority of our participating GPs and CPs were interested to participate in the regular GP-CP antimicrobial pharmacotherapy group meetings, but an attitudinal divergence was significant (GPs vs. CPs; 54.9% vs. 82.5%; p<0.001)”

“Less than a quarter of our participants with no significant difference between GPs and CPs (p<0.116) used the patient information leaflets to treat infections, reflecting a limited provision of using leaflets.”

“Similarly, the uptake of point-of-care tests by GPs and CPs was below 20% with no interprofessional difference (p<0.784)”

“The use of the Therapeutic Guidelines: Antibiotics24 was not a common practice by CPs; significantly lower than GPs (CP vs GP; 45.5% vs 83.2%; p<0.0001)”

Reviewer 2 Report

Saha et al. conducted a nationwide survey involving both general practitioners and community pharmacists to implement antimicrobial stewardship programs in Australia. It is an interesting piece of work and suitable to publish in antibiotics.  The authors noted that CPs is interested in AMS training than general practitioners. Furthermore, the inter collaboration between CPs and GPs will help implement GPPAS strategies and similarly, it can be applied in other nations. In general, the manuscript is written well and good.

There can few points be included in the conclusion concerning where exactly need strong collaboration between CPs and GPs to minimize the risk of future Antimicrobial Resistance.

Did the authors survey the use of last-resort antibiotics? Is there any indiscrimination and I did not find such information. Moreover, what kinds of antibiotics are predominantly prescribed by GPs.

Author Response

Saha et al. conducted a nationwide survey involving both general practitioners and community pharmacists to implement antimicrobial stewardship programs in Australia. It is an interesting piece of work and suitable to publish in antibiotics.  The authors noted that CPs is interested in AMS training than general practitioners. Furthermore, the inter collaboration between CPs and GPs will help implement GPPAS strategies and similarly, it can be applied in other nations. In general, the manuscript is written well and good.

Thank you for your positive comments.

There can few points be included in the conclusion concerning where exactly need strong collaboration between CPs and GPs to minimize the risk of future Antimicrobial Resistance.

We have included few points in the conclusion section of the manuscript as follows.

"There is a need for GPs and CPs to recognise their interprofessional roles in identifying patients who truly need antimicrobials, and improving the choice, dose, and duration of antimicrobials to reduce avoidable AMR. The arrangement of health system structure and a policy improving the GP-pharmacy collaborative practice agreement would support the development of a GP-CP collaborative antimicrobial care model in Australia."

Did the authors survey the use of last-resort antibiotics? Is there any indiscrimination and I did not find such information? Moreover, what kinds of antibiotics are predominantly prescribed by GPs.

This topic was beyond the scope of our survey questions we designed.

Round 2

Reviewer 1 Report

Thank you for your prompt response to my original suggestions. I fully agree with the changes you have made. In my opinion the manuscript now reflects the results of your survey and provides a solid blueprint for addressing the challenges of instituting meaningful collaborative community-based stewardship programs between GPs and CPs.